# Township, County Town, Metropolitan Area, or Foreign Cities? Evidence from House Purchases by Rural Households in China

Chengxiang Wang [1,2], Zehua Pang [2,3] and Chang Gyu Choi [1,*]

1   Urban Design Analysis Lab, Graduate School of Urban Studies, Hanyang University, Seoul 04763, Republic of Korea; liutanghe@hanyang.ac.kr
2   Institute of Land and Urban-Rural Planning, Huaiyin Normal University, Huai'an 223300, China; m210872j1008@stu.hut.edu.cn
3   College of Urban and Environmental Sciences, Hunan University of Technology, Zhuzhou 412007, China
*   Correspondence: cgchoi@hanyang.ac.kr

**Abstract:** In the context of China's New Urbanization Strategy, it is of great practical significance to study rural–urban population migration from the perspective of house purchases by rural households. In this paper, the Huai'an Rural Survey Dataset (872,414 households) was used to study the heterogeneity of rural households' house purchases in different classes of urban destinations, and its influencing factors were analyzed with GeoDetector. The results show that the urban house purchase destinations preferred by farmers were county towns, townships, foreign cities, and metropolitan areas, indicating that in situ urbanization has become the main path of urbanization for farmers in Huai'an. Among the environmental influencing factors, the rural environment had the greatest influence on house purchases locally (in the township and county town), and this influence decreased with the outward shift of house purchase destinations. The housing environment, the settlement environment, and the population and family environment were the main environmental impact elements. The natural environment and the policy environment had little influence on the house-purchasing behavior of farmers, and the location environment was critical in exotic locations (metropolitan areas and foreign cities). Therefore, this paper argues that a higher demand for housing is growing in China's less developed rural areas, creating a situation in which the metropolitan area is the core and the county town is the main contributor. In terms of policy improvements, it is important to pay more attention to small cities such as counties and to offer housing concessions and welfare to "new citizens" from rural areas, as well as to significantly improve the housing, earnings, and public service environment for those who prefer to stay in the countryside.

**Keywords:** rural households; urban house purchase; GeoDetector; spatial pattern; influencing factors; Huai'an City

## 1. Introduction

Since the late 1980s, China has experienced the massive migration of peasants into the cities, called the "migrant workers tide" [1,2], which denotes the huge urban–rural split and the eagerness of farmers to improve their livelihoods. According to the Bulletin of the Seventh National Census of China (2020), compared with the Sixth National Census of China (2010), the urban population increased by 236 million and the rural population decreased by 164 million [3]. Moreover, there is a floating population consisting of 376 million people, most of whom are from rural areas [3,4]. The migration behavior of farmers shows diversification, complexity, and magnitude. After decades of rural–urban ferrying, urban cognition, and adaptation, the rural population has gradually shifted to purchasing properties and settling down permanently in cities along with their families for stability [5,6]. "Buying a house in the city" has become a consensus in the countryside and has served as a microcosm of China's rapid urbanization.

In the 21st century, with the abolition of the custody and repatriation system (CRS) and the loosening of restrictions on urban settlements, many cities have made it possible to achieve "citizenship" by purchasing a house. At the same time, market mechanisms have begun to influence farmers' migration [7]. The purchase of houses can contribute to the rapid development of the real estate, service, and labor markets, which can significantly increase land revenue and promote the modernization of public infrastructure and landscapes. Therefore, urban governments have introduced many policies to encourage rural households to purchase houses [8]. However, excessive migration has caused urban problems, such as traffic congestion, housing tensions, and pollution [9,10]. Rural areas have also suffered a systematic decline, including aging, industrial stagnation, and environmental degradation [11]. Since 2017, the Chinese government has been implementing a comprehensive rural revitalization strategy, and one of its priorities is to prevent excessive population loss, as in the 2018 No. 1 Central Document [12] and the "Opinions on Comprehensively Promoting Rural Revitalization and Modernization" [13]. In short, achieving a balance between flowing into cities, staying in cities, and returning to the countryside has been approved by administrators and scholars, and the regulation of rural households' purchasing behavior has become a vital approach.

Ideally, farmers would prefer cities with better public services, employment opportunities, and living environments [14–16]. In reality, however, many factors prevent them from purchasing houses in target cities [8,17]. The push of the moving-out place, the pull of the moving-in place, and personal factors are all influencing rural–urban migration. However, previous studies have mainly concentrated on the destination (cities) and the migrants [18–22], ignoring the origin (rural) environment. China's 14th Five-Year Plan for National Economic and Social Development proposed "promoting the New Urbanization Strategy with people at its core, and promoting the coordinated development of large, medium, small cities, and small towns" [23]. Therefore, it is important to identify the heterogeneity of rural households' house purchase characteristics to optimize the layout of public facilities and promote the coordinated development of urban and rural areas.

On this basis, we used house purchase data from the full-sample countryside (872,414 households) to study the behavior of rural households. First, we drew on and refined amphibious farmers, which is of contemporary significance in China. In addition, our research is bottom-up, taking farmers and the rural environment as the foothold, which expands the research perspective to the best of our knowledge. The most important aspect is that house purchase destinations were classified into four groups: township, county town (or district), metropolitan area, and foreign cities. The spatial patterns and driving factors were analyzed at the micro-scale (1307 administrative villages) for solving housing problems in rural–urban migration and in situ urbanization.

The remaining sections are as follows. First, we reviewed the literature on rural–urban migration and identified the essentials of inequality and the characteristics of migration in China in Section 2. In Section 3, we present the data sources, research methods, and frameworks. This is followed by an empirical analysis that examines the spatial heterogeneity and drivers of house purchasing in Section 4. The final section, Section 5, includes the conclusions and a discussion.

## 2. Literature Review

### 2.1. Homeownership: Environmental Inequality

The phenomenon of farmers' house purchasing in cities is essentially the result of migration. The Growth Pole Theory explains the inequalities that emerge under rapid industrialization and urbanization [24]: migrants have always flowed from underdeveloped regions (e.g., villages and towns) to developed growth poles (large cities), especially in developing countries [25,26]. The origin of this phenomenon is the surplus labor force in rural areas seeking opportunities such as higher incomes, better public services, and greater status in cities [18]. For example, Seoul, the only central city in Korea, has become the primary destination for rural and other urban residents [14]; in China, a large number of

people moved into the Pearl River Delta region and the Yangtze River Delta region [15,17]. Although there was an out-migration of citizens due to suburbanization in developed countries, such as the USA, there are still only a few cases in the post-urbanization era [27].

Some theories, such as Lee's "Push–Pull Theory", provide an authoritative and widely applicable approach to population migration [28,29]. Other theories, such as Neoclassical Economics, argue that differences in wage performance arising from employment imbalances are the main cause of population migration [30]; the New Economic of Migration Theory views households as subjects to maximize earnings [31]. In short, the inequality between urban and rural areas is the main reason for farmers' inflow to cities, as reflected in the high vitality of cities and the deadness of the countryside.

Scholars have tried to identify the multidimensional influences on farmers' migration to cities. Numerous studies from an in-migration perspective have found that higher salaries, more comfortable housing, more accessible transportation, higher-quality public services, and greater governability in cities are the main attractions for immigrants [14,16,18,20]. Moreover, research on individual farmers has revealed that economic situations, rural housing conditions, household aging, education attainment, and social relationships have a greater impact on willingness to move [17,21,22,32–35]. On the negative side, housing prices and social exclusion can be barriers to the urban integration of farmers [36–38]. However, few studies have focused on the multi-category environmental elements of rural habitats (out-migration), such as the industrial environment, the employment environment, and the ecological environment. Actually, the spatial difference between urban and rural areas, i.e., environmental inequality, is the primary reason for "pushing" farmers out of the countryside.

### 2.2. From Village to City: The Story of China's Farmers

The issue of China's population migration and house purchasing has Chinese characteristics. In the late 1980s, the Chinese government encouraged peasants to move into cities to meet labor shortages in emerging industries and services [2], while peasants were also eager for cities to help improve their quality of life. Consequently, there was an influx of people into China's megacities, such as Shanghai, Beijing, and provincial capitals [15,17,39]. As the market economy flourished in China, real estate became an important tool for the local governments to increase revenue and promote urbanization [40], thus formulating a series of policies, such as compulsory education zoning [41]. Under such circumstances, the purchase of houses in cities by farmers has changed from an individual intention to a "collective willingness of that times". In addition, the Chinese cultural perception of "settling down to start a family" is a special factor that motivates buying instead of renting a house [42].

China's long-standing household registration system (hukou) divides urban residents into privileged urban aborigines and disadvantaged rural newcomers [43]. As a result, a new group was created, namely those whose hukou remained rural, but in fact, lived in the city as citizens. Some families struggle to change their household registration and become outright urban families, a significant contributing factor limiting their children's education and social insurance [44,45]. However, more rural migrants have difficulty changing their hukou and benefit from citizenship [44,46]. Despite the fact that cities are so attractive, some rural families refuse to move their registered residences to the cities and instead become "amphibious farmers" [47]. There are three main reasons for this: the first is that they still possess rural assets [22]; the second is that they have their socialization in the countryside [48]; and third, traditional Chinese sentiments, such as "returning to one's roots" and "homeland is hard to leave", become ties to their hometown, especially for the elderly [49,50].

As discussed above, the spatial heterogeneity of farmers' migration destinations has become more pronounced in recent years. Instead of heading to megacities, many farmers have been moving to local counties or municipalities nearby, which is called in situ urbanization and is widely perceived as an essential path to bridging the urban–rural

gap and revitalizing the countryside [51,52]. Arguably, in situ urbanization and relocation urbanization exist simultaneously, with different rural migrants often attracted to various urban destinations [53]. However, most studies have investigated rural migrants in different cities to derive multifactorial differences between large, medium, and small cities [5,15,54]. Only a few scholars have studied farmers' migration intentions in rural areas, and the sample sizes are generally not large [55,56]. In short, there is a paucity of research on the environment of rural migration sites in China. Therefore, a disaggregated survey from the perspective of farmers and rural areas is necessary and should incorporate multiple factors of individuals, families, settlements, and rural living environments. The reason is that, unlike potential urban destinations, the original rural living environment is authentic and objective.

## 3. Data and Methods

### 3.1. Study Area and Data

In this paper, the study area is the countryside of Huai'an City, which is located in the north of Jiangsu Province, a developed province in eastern China (Figure 1). However, the three sub-regions of Jiangsu Province, Southern Jiangsu, Central Jiangsu, and Northern Jiangsu, have faced long-term imbalanced development [57]. Since the rise of the Southern Jiangsu Model in the 1980s, the rapidly developing manufacturing industry has created massive employment with substantial wages, leading to a gap between Southern and Northern Jiangsu. In addition, the proximity to Shanghai provides tremendous development opportunities, leading to long-term population loss and unsustainability in Northern Jiangsu, especially in the rural areas [58].

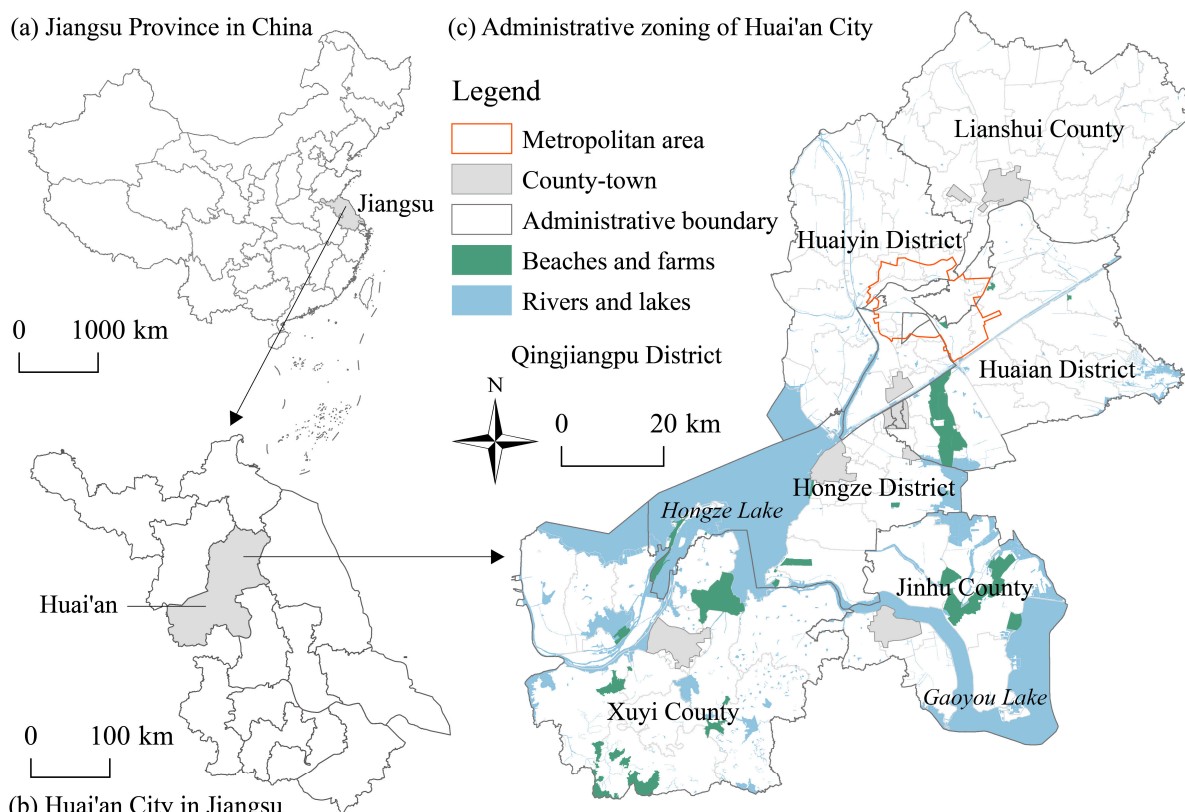

**Figure 1.** The study area.

Huai'an is a geographic center city in Northern Jiangsu with strong representativeness. In 2021, Huai'an had a registered rural population of 2,254,300 and a resident population of 1,541,600, with a rural population outflow rate of 31.62% [59]. Long-term population loss has brought many problems to rural Huai'an, such as a significant lack of vitality, aging,

and lagging industrial development. As per our 2019 survey data, the vacant housing rate in rural Huai'an was 17.13%. With the implementation of a series of revitalization plans for Northern Jiangsu, regional inequality and polarization have steadily declined [57]. In particular, the comprehensive strength of the metropolitan area has increased and absorbed a great number of local farmers. However, Huai'an is still characterized by urban strength and rural weakness, and most villages and towns still do not possess the strength to keep farmers [60]. Therefore, we believe that the rural area of Huai'an is a good sample for studying the purchasing behavior of amphibious farmers.

The data used in this paper are from the full-sample household survey data in rural areas released by the Huai'an Bureau of Statistics in 2019, including a total of 872,414 households. The dataset used administrative villages rather than individuals as the statistical unit, including data on the housing situation, population, natural conditions, public services, ecological environment, and other aspects. As the study subjects, we selected 1307 administrative villages after excluding towns, communities, and outliers. Meanwhile, we compiled some data from the yearbook as research variables [59]. The geospatial data are derived from The Third National Land Survey, and some points of interest (POI), such as school and government sites, were obtained from AMAP [61].

### 3.2. Variables and Methods

#### 3.2.1. Dependent Variables: Explaining the Destinations

We defined the dependent variable as the urban house purchase rate of amphibious farmers in administrative villages, i.e., the number of urban house purchases as a proportion of all rural households. On the basis of China's administrative division system, our dependent variable group included the house purchase rate in the township, the house purchase rate in the county town, the house purchase rate in the metropolitan area, and the house purchase rate in foreign cities.

First, the house purchase rate in the township represents the proportion of rural households that purchased in the built-up area of the town ("ZhenQu") to which a village belongs. Second, the house purchase rate in the county town represents the proportion of rural households that purchased in the central urban area of the county ("XianCheng") to which a village belongs. Third, the house purchase rate in the metropolitan area represents the proportion of rural households who purchased in the central urban area of Huai'an City ("ShiQu"). Fourth, the house purchase rate in the foreign cities represents the proportion of rural households who purchased in any other city ("WaiDi"), either in Shanghai or Nanjing. For example, in Hongxing Village, Zhuma Town, Lianshui County, and Huai'an City, the purchase rate in a township only includes the built-up area of Zhuma Town, and its county town is the only central urban area of Lianshui County. We believe that those definitions are consistent with the concept of in situ urbanization.

#### 3.2.2. Independent Variables: Measuring the Rural Environment

Fewer studies have been conducted on the factors influencing the environment at the village scale, but studies on migration can be drawn on. Combining previous studies and field surveys in rural Huai'an [60], this paper classifies the influence of village geographical environment on rural house purchase into eight categories (Table 1): natural environment, settlement environment, housing environment, economic environment, population and family environment, location environment, public service environment, and policy environment.

The natural environment includes two variables, *slope* and *pollution degree*, which impact the comfort and security of rural living [60]. We predict that the worse the natural conditions are, the more likely it is that people will purchase houses in the cities. Of these, mountainous landscapes in plain areas may be valued and used for tourism development, which may increase the well-being of farmers and reduce their willingness to migrate. However, pollutants have a negative impact on people in any area and generally drive people away.

**Table 1.** Indicator selection.

| Indicators | Definition | Min | Mean | Max |
|---|---|---|---|---|
| **Natural Environment** | | | | |
| $X_1$ *slope* | Average degree of slope of the administrative village (°) | 0.01 | 0.67 | 6.72 |
| $X_2$ *pollution degree* | Sum of pollution sites in the administrative village (pcs) | 0 | 6.40 | 78.00 |
| **Settlement Environment** | | | | |
| $X_3$ *road density* | Road density of the administrative village (km/km$^2$) | 1.17 | 21.51 | 99.40 |
| $X_4$ *per capita arable land* | Ratio of arable land area to number of households (acre) | 0.01 | 2.06 | 111.72 |
| $X_5$ *size of natural villages* | Average number of households per natural village (pcs) | 2.60 | 66.46 | 473.00 |
| $X_6$ *settlement connectivity* | Average distance between natural villages (km) | 0.03 | 0.99 | 3.15 |
| **Housing Environment** | | | | |
| $X_7$ *proportion of buildings* | Ratio of number of buildings ($\geq$2 stories) to households (%) | 0.96 | 41.78 | 96.96 |
| $X_8$ *per family homestead size* | Ratio of total homestead area to households (acre) | 0.01 | 0.67 | 4.69 |
| $X_9$ *per capita living area* | Ratio of total housing area to population (m$^2$) | 12.15 | 39.77 | 96.34 |
| $X_{10}$ *housing quality* | Proportion of houses with acceptable quality (%) | 5.94 | 73.61 | 100.00 |
| **Economic Environment** | | | | |
| $X_{11}$ *per household income* | Average annual income of resident households (CNY 10,000) | 1.20 | 5.97 | 14.91 |
| $X_{12}$ *number of enterprises* | Number of registered enterprises (pcs) | 0.00 | 9.05 | 256.00 |
| $X_{13}$ *fiscal revenue* | Annual fiscal revenue of its town (CNY million) | 11.93 | 127.37 | 543.72 |
| $X_{14}$ *proportion of poor families* | Proportion of registered low-income households (%) | 0.00 | 13.81 | 54.53 |
| **Population and Family Environment** | | | | |
| $X_{15}$ *dependency ratio* | Average number of elderly and children to support per worker | 0.44 | 0.93 | 1.70 |
| $X_{16}$ *household density* | Ratio of permanently settled households to area (pcs/km$^2$) | 1.00 | 111.18 | 372.64 |
| $X_{17}$ *proportion of leavers* | Proportion of whole families leaving frequently (%) | 0 | 12.02 | 58.40 |
| $X_{18}$ *proportion of vacant house* | Proportion of vacant (abandoned) buildings (%) | 0 | 18.05 | 81.03 |
| **Location Environment** | | | | |
| $X_{19}$ *distance to metro area* | Travel time by car to the center of Huai'an City (min) | 8.46 | 49.66 | 113.77 |
| $X_{20}$ *distance to county town* | Travel time by car to its county government site (min) | 1.53 | 23.87 | 53.51 |
| $X_{21}$ *distance to traffic station* | Travel time by car to the nearest transport station (min) | 0.01 | 5.40 | 20.98 |
| $X_{22}$ *distance to township* | Travel time by electric bike to its town government site (min) | 0.01 | 17.12 | 63.87 |
| **Public Service Environment** | | | | |
| $X_{23}$ *commercial development* | Number of shops (pcs) | 0 | 59.86 | 1556 |
| $X_{24}$ *education accessibility* | Travel time by car to the nearest high school (min) | 0.01 | 18.62 | 50.29 |
| $X_{25}$ *healthcare accessibility* | Travel time by car to the nearest secondary hospital (min) | 0.10 | 19.46 | 53.45 |
| $X_{26}$ *kindergarten accessibility* | Travel time by electric bike to the nearest kindergarten (min) | 0.01 | 9.09 | 44.59 |
| **Policy Environment** | | | | |
| $X_{27}$ *proportion of policy village* | Proportion of natural villages with development policies (%) | 0 | 6.96 | 100 |
| $X_{28}$ *proportion of reserve* | Proportion of natural villages with ecological reserves (%) | 0 | 9.33 | 100 |

We incorporate *road density*, *per capita arable land*, *size of natural villages*, and *settlement connectivity* into the settlement environment. The agglomeration effect of villages can create a sense of belonging among farmers, and usually, more tightly connected and larger settlements are more likely to keep farmers living locally [20,35].

What this paper discusses is the heterogeneity of rural residents' house purchases in different destinations, with the housing environment in the original villages being the main influencing factor [22,60]. Rural households' homestead is a unique policy land in rural China, and farmers cannot purchase or expand it at will, nor can they build houses beyond the standard [34]. Therefore, *per family homestead size* is used here as a specificity indicator. With the steady increase in farmers' earnings, the demand for housing size and quality has increased, so *proportion of buildings*, *per capita living area*, and *housing quality* were selected for analysis.

The next indicator is the economic environment of the village, which is the decisive factor [18,20]. The *number of enterprises* in the village and the *fiscal revenue* of the town are chosen to represent the overall economic performance of the village, which reflects the manufacturing capacity of the village and the employment it can provide. This reflects the level of manufacturing in the village and the jobs it can provide. In addition, *per household income* and *proportion of poor families* reflect individual and household economic conditions [19].

The population and family environment is an important factor that influences house-purchasing behavior [54]. Chinese traditional culture emphasizes "respecting the old and caring for the young", and older parents and children can play a role in the settlement of prime-age farmers [49]. Therefore, we use *dependency ratio* as its indicator. Additionally, *household density*, *proportion of leavers*, and *proportion of vacant house* reflect the behavioral choices of the rural family.

The impact of the location environment is objective and significant and has interactive effects with other environmental elements [18]. We selected *distance to township*, *distance to county town*, and *distance to metro area* as indicators of location advantage. We also added an auxiliary indicator, that is, *distance to traffic station*, including highway entrances and exits, transportation hubs, etc.

The public service environment cannot be ignored. The gap between urban and rural areas is mainly reflected in education, medical care, transportation, and other aspects that the countryside lacks [16]. We chose *commercial development*, *education accessibility*, and *kindergarten accessibility* to reflect the accessibility of public services [7]. In particular, children's education is the most important to Chinese families, and the quality of education can have a great impact on the choice of location. Another indicator, *healthcare accessibility*, needs to be included in our framework, and we use the travel time to the nearest secondary hospital as the data.

The last is the policy environment [60]. The Chinese Ministry of Natural Resources clearly classifies the types of villages into four categories: clustering and upgrading, suburban integration, characteristic protection, and relocation and removal of villages. We took the proportion of the top three types of natural villages among administrative villages as this indicator, that is, the *proportion of policy village*. The study also found that ecological protection areas impose policy constraints on village development but increase the preservation of cultural characteristics. Therefore, we used *proportion of reserve* as an indicator.

### 3.2.3. Methods

In this paper, we used spatial autocorrelation analysis to explore the geographical differences in the urban house purchase behavior of amphibious farmers [62,63]. The Global Moran's *I* was used to examine the spatial autocorrelation characteristics and thus determine whether there is a spatial measurement significance in farmers' behavior when purchasing houses in cities. The formula is as follows:

$$\text{Moran's } I = \frac{n}{\sum_{i=1}^{n} \sum_{j=1}^{n} W_{ij}} \times \frac{\sum_{i=1}^{n} \sum_{j=1}^{n} W_{ij} \left(y_i - \bar{y}\right) \left(y_j - \bar{y}\right)}{\sum_{i=1}^{n} \left(y_i - \bar{y}\right)^2} \tag{1}$$

where $n$ is the total number of village units; $y_i$ and $y_j$ denote the attribute value (purchase rate) of the $i$th village units and the $j$th village units, respectively; $W_{ij}$ is the spatial weight matrix; and $\bar{y}$ is the average value of all units. The larger the absolute value of Moran's $I$, the stronger the global spatial autocorrelation.

Another analysis type is local spatial autocorrelation analysis. Hot spot analysis (Getis–Ord $G_i^*$) is used to represent the local agglomeration of house purchase rates among cities and is divided into cold-spot and hot-spot clusters. The formula is as follows:

$$Gi* = \frac{\sum_{j=1}^{n} W_{ij}y_j - \frac{\sum_{j=1}^{n} y_j}{n}\sum_{j=1}^{n} W_{ij}}{\sqrt{\frac{\sum_{j=1}^{n} y_j^2}{n} - \left(\frac{\sum_{j=1}^{n} y_j}{n}\right)^2}\sqrt{\frac{n\sum_{j=1}^{n} W_{ij}^2 - (\sum_{j=1}^{n} W_{ij})^2}{n-1}}} \tag{2}$$

The analysis results were generally normalized and recorded as $Z$ ($G_i^*$). The larger the value of $Z$ ($G_i^*$), the greater the level of cold-spot (low–low clustering) and hot-spot (high–high clustering) areas.

Previous studies have commonly used OLS regression to analyze the influencing factors, but OLS can only estimate the correlation of parameters at the global level. The comprehensive framework constructed in this paper has more environmental elements, and traditional methods may have a co-integration that affects the conclusions. Therefore, this paper used GeoDetector, a method for detecting geospatial anisotropy without linear assumptions [63,64]. Basically, the study area is assumed to be divided into sub-regions, and spatial heterogeneity exists if the sum of the variances of the sub-regions is smaller than the total variance of the area. If the spatial distribution of the two variables tends to be the same, then there is a statistical correlation between them. The first sub-method we use is the factor detector, which uses the $q$ statistic to detect the extent to which a given environmental variable $X$ explains the dependent variable, house purchasing. The formula is as follows:

$$q = \frac{1 - \sum_{h=1}^{L} N_h\sigma_h^2}{N_\sigma^2} \tag{3}$$

where $h$ represents the strata of any variable; $N_h$ and $N$ are the number of units in the layer $h$ and the whole area, respectively; $\sigma_h^2$ and $\sigma^2$ are the variances of the dependent variable of the layer $h$ and the whole area, respectively; and $0 \leq q \leq 1$. The larger the $q$ statistic, the stronger the explanatory power of the environmental variable for the dependent variable, and vice versa.

Another sub-detector is the interaction detector, which can detect the degree of interaction between any two factors. In our study, it is essential to reinforce the role of the purchase factor through an interactive approach. We can use the interaction detector to assess whether $X_1$ and $X_2$ increase the explanatory power of the dependent variable when acting together; its parallel implication is that the two variables are multiplied. When $X_1$ and $X_2$ interact, the $q$ statistic is denoted as $q$ ($X_1 \cap X_2$). First, the $q$ statistics of the effects of the two factors $X_1$ and $X_2$ on the dependent variable are calculated separately, denoted by $q$ ($X_1$) and $q$ ($X_2$). $q$ ($X_1 \cap X_2$) is then compared with $q$ ($X_1$) and $q$ ($X_2$), and their relationships are classified into the following five categories: If $q$ ($X_1 \cap X_2$) < Min ($q$ ($X_1$), $q$ ($X_2$)), it is called Weaken-nonlinear. If Min ($q$ ($X_1$), $q$ ($X_2$)) < $q$ ($X_1 \cap X_2$) < Max ($q$ ($X_1$), $q$ ($X_2$)), it is called Single Weaken-nonlinear. If $q$ ($X_1 \cap X_2$) > Max ($q$ ($X_1$), $q$ ($X_2$)), it is called Enhance-bifactor. If $q$ ($X_1 \cap X_2$) = $q$ ($X_1$) + $q$ ($X_2$), it is called Independent. If $q$ ($X_1 \cap X_2$) > $q$ ($X_1$) + $q$ ($X_2$), it is called Enhance-nonlinear. Both Weaken-nonlinear and Single Weaken-nonlinear indicate the existence of a mutually antagonistic relationship between them, which together will produce a weaker effect on the dependent variable than if they were acting separately. The other two interactions, Enhance-bifactor and Enhance-nonlinear, both indicate that they work together to make a stronger contribution to the dependent variable. However, GeoDetector also has a drawback in that it does not show the positive and negative characteristics of the influencing factors and therefore uses the OLS model for assistance.

## 4. Results

### *4.1. Destinations for Urban Housing Purchases*

#### 4.1.1. Descriptive Statistics

Table 2 presents descriptive statistics of the different destinations where rural households purchased houses. First, as far as we know, 44.66% of rural households purchased houses in cities and towns. When the administrative village was used as the study unit, the average urban house purchase ratio was 45.96%, indicating that our study population is an essential and non-negligible group. Of these, the lowest percentage and the highest percentage of houses purchased by villagers were 0.16% and 99.79%, respectively, which reflects the great variability among villages.

**Table 2.** Descriptive statistics of urban house purchase rate in different destinations.

|  | **Min** | **Median** | **Max** | **Mean** | **CV** |
|---|---|---|---|---|---|
| All | 0.16% | 47.24% | 99.79% | 45.96% | 0.400 |
| In township | 0.00% | 9.11% | 99.79% | 14.63% | 1.110 |
| In county town | 0.00% | 15.08% | 63.38% | 18.78% | 0.741 |
| In metropolitan area | 0.00% | 3.19% | 47.88% | 4.82% | 1.207 |
| In foreign cities | 0.00% | 6.21% | 55.88% | 7.74% | 0.837 |

There are large statistical differences among the four house purchase destinations. In terms of mean value, the destinations for house purchases are, from most to least, the county town, the township, foreign cities, and the metropolitan area. The fact that the maximum number of houses were purchased in a county town is consistent with China's promotion of in situ urbanization with the county as the carrier (18.78%). The second-highest number of houses were purchased in the home township (14.63%), but the coefficient of variation (CV) and the maximum values of this indicator are 1.11 and 99.79%, respectively, indicating that the attractiveness and development of townships in Huai'an are extremely uneven. This is because there are few well-developed towns in Huai'an that inhibit population outflow due to the prominence of manufacturing, and most towns are poorly developed. Finally, the average percentage of house purchases in foreign cities was 7.74%, which is more than the percentage of houses purchased in the Huai'an metropolitan area (4.82%), indicating that Huai'an is less attractive than outside cities.

#### 4.1.2. Spatial Pattern and Association Feature

Figure 2 presents the spatial pattern of urban house purchases in different destinations. The villages with high purchase rates in the township were concentrated in areas with strong development levels, such as Maba Town in Xuyi County and Gaogou Town in Lianshui County (Figure 2a). Among them, Gaogou Town had one alcohol company listed on the Shanghai Stock Exchange, while Huai'an had only two A-share-listed companies in 2019. By contrast, Maba Town is located near two highways and is easily accessible from the outside. Figure 2b shows that high house purchase rates in the county town were concentrated around the central built-up areas of Jinhu County, Hongze District, and Lianshui County, which are characterized by a single-core urban system. Figure 2c reveals that rural farmers with better accessibility to metropolitan areas, such as the periphery of Huaiyin District, Qingjiangpu District, and Huai'an District, tended to purchase houses in metropolitan areas. Finally, Figure 2d exposes the tendency of rural households at the edge of Huai'an to purchase houses in foreign cities, such as Huanghuatang Town in Xuyi County and Pingqiao Town in Huai'an District bordering the provincial capital Nanjing.

Figure 3 and Table 3 reveal the results of global and local spatial association features, respectively. Table 3 shows that Moran's *I* is greater than 0 and higher for all destinations, $Z > 0$, $p = 0.000 < 0.001$, indicating that farmers' urban house purchase behavior presents a significant global spatial autocorrelation in rural Huai'an. Local autocorrelation (Figure 3)

shows that the hot and cold spots of house purchases in different destinations are evident and present essentially similar findings to the spatial pattern.

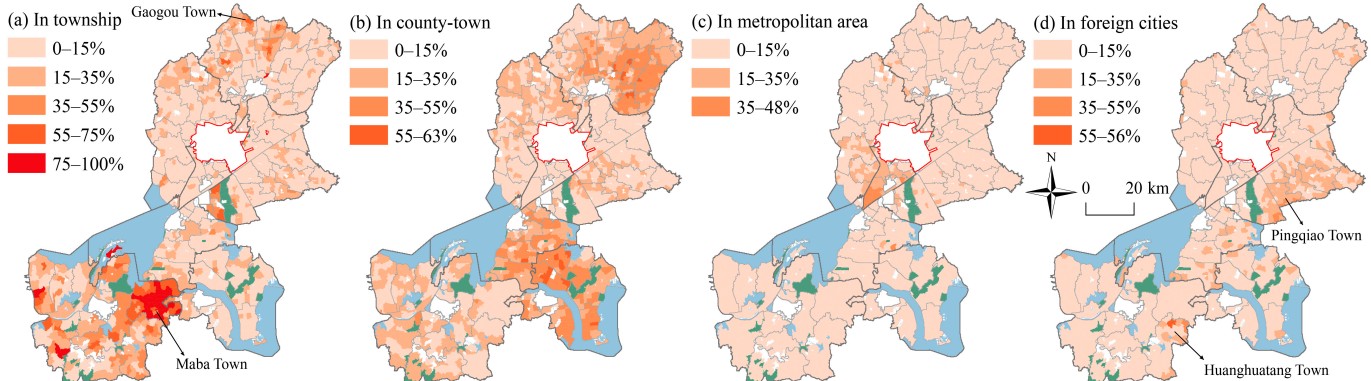

**Figure 2.** Spatial pattern of housing purchases by rural households in different destinations.

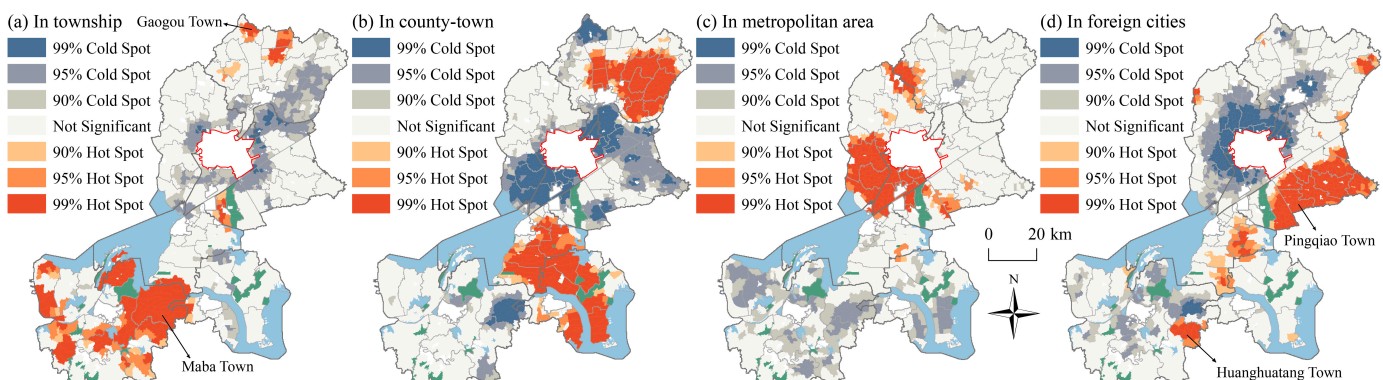

**Figure 3.** Local autocorrelation of housing purchase rates by rural households in different destinations.

**Table 3.** Results of global spatial autocorrelation of housing purchases rates.

|  | Moran's *I* | Z | *p* |
|---|---|---|---|
| All | 0.528 | 41.823 | 0.000 |
| In township | 0.410 | 32.517 | 0.000 |
| In county town | 0.698 | 55.275 | 0.000 |
| In metropolitan area | 0.525 | 41.727 | 0.000 |
| In foreign cities | 0.647 | 51.315 | 0.000 |

*4.2. Determinants of Housing Purchases*

4.2.1. Comparison of Different Destinations

Table 4 shows the magnitude of the drivers for the factor detector. First, we analyzed it from the perspective of different house purchase destinations. Compared to the other three destinations, purchasing a house in the township (Model 1) was influenced to a more drastic degree by various environmental factors. Of these, *proportion of buildings* ($X_7$) had the largest influence, reaching a value of −0.243, and the influence of *per capita living area* ($X_9$) was ranked fourth, indicating that farmers who purchased in the local area were more concerned about the improvement of the basic demand for living quality. The secondary factor was population, confirming the fact that population mobility and house purchasing were linked. The impact of each factor on purchasing a house in the county town (Model 2) was slightly lower than that in the township (Model 1). Three factors had a *q* greater than 0.1, and the largest factor was still *proportion of buildings* ($X_7$). The second factor was *distance to metro area* ($X_{19}$), which indicates that farmers who purchased in the county town

were satisfied with the development of their county. The next one was in the metropolitan area (Model 3), where only two indicators had a $q$ greater than 0.1, indicating that those amphibious farmers were less influenced by environmental factors in the countryside and were more attracted by the city. From the farmers' perspective, entering a metropolis means disconnecting from the network of rural relationships. Model 4 presents the factors influencing the purchase of a house in a foreign city, where the most influential one was still *distance to metro area* ($X_{19}$) and the third one was *distance to county town* ($X_{20}$), which reveals that farmers who purchased outside were troubled by the location. Fieldwork also demonstrates that some of the most marginalized rural areas in Huai'an may be attracted to Nanjing. In addition, *fiscal revenue* ($X_{13}$) becomes the second-most-important indicator. Usually, more capital is required to purchase in a larger city in China [15].

**Table 4.** Results of factor detector.

| | Model 1 | Model 2 | Model 3 | Model 4 |
|---|---|---|---|---|
| | **In Township** | **In County Town** | **In Metropolitan Area** | **In Foreign Cities** |
| **Natural Environment** | | | | |
| $X_1$ *slope* | 0.029 * [1] | −0.030 ** | 0.021 *** | −0.027 *** |
| $X_2$ *pollution degree* | 0.010 *** | −0.026 *** | 0.006 | −0.011 |
| **Settlement Environment** | | | | |
| $X_3$ *road density* | −0.082 *** | 0.045 *** | −0.053 *** | −0.022 *** |
| $X_4$ *per capita arable land* | 0.154 ***(H) | −0.096 *** | −0.044 *** | −0.020 * |
| $X_5$ *size of natural village* | 0.090 *** | −0.022 ** | −0.006 | −0.015 |
| $X_6$ *settlement connectivity* | −0.041 *** | 0.033 *** | 0.019 ** | 0.009 |
| **Housing Environment** | | | | |
| $X_7$ *proportion of buildings* | −0.243 ***(H) | −0.220 ***(H) | 0.106 ***(H) | 0.072 *** |
| $X_8$ *per family homestead size* | 0.134 ***(H) | −0.049 *** | 0.062 *** | −0.032 *** |
| $X_9$ *per capita living area* | −0.169 ***(H) | −0.088 *** | 0.077 *** | −0.103 ***(H) |
| $X_{10}$ *housing quality* | 0.033 *** | 0.021 ** | 0.034 *** | −0.041 *** |
| **Economic Environment** | | | | |
| $X_{11}$ *per household income* | −0.020 ** | 0.018 * | 0.007 | 0.072 *** |
| $X_{12}$ *number of enterprises* | −0.022 * | 0.059 *** | −0.018 | 0.024 *** |
| $X_{13}$ *fiscal revenue* | 0.130 ***(H) | −0.038 *** | 0.046 *** | 0.117 ***(H) |
| $X_{14}$ *proportion of poor families* | 0.016 * | −0.047 *** | −0.018 * | 0.053 *** |
| **Population and Family Environment** | | | | |
| $X_{15}$ *dependency ratio* | −0.038 *** | 0.015 * | −0.011 | −0.006 |
| $X_{16}$ *household density* | −0.205 ***(H) | 0.121 ***(H) | −0.034 *** | 0.027 *** |
| $X_{17}$ *proportion of frequent leavers* | −0.031 *** | 0.041 *** | 0.008 | −0.045 *** |
| $X_{18}$ *proportion of vacant houses* | 0.207 ***(H) | −0.069 *** | 0.026 *** | 0.061 *** |
| **Location Environment** | | | | |
| $X_{19}$ *distance to metro area* | 0.107 ***(H) | 0.159 ***(H) | −0.236 ***(H) | −0.133 ***(H) |
| $X_{20}$ *distance to county town* | 0.023 *** | −0.025 *** | −0.019 ** | 0.111 ***(H) |
| $X_{21}$ *distance to traffic station* | −0.011 | 0.034 *** | 0.023 *** | 0.047 *** |
| $X_{22}$ *distance to township* | 0.076 *** | −0.023 ** | −0.020 ** | 0.022 ** |

**Table 4.** *Cont.*

| | Model 1 | Model 2 | Model 3 | Model 4 |
| --- | --- | --- | --- | --- |
| | In Township | In County Town | In Metropolitan Area | In Foreign Cities |
| **Public Service Environment** | | | | |
| $X_{23}$ *commercial development* | 0.035 ** | −0.034 *** | 0.047 *** | −0.063 *** |
| $X_{24}$ *education accessibility* | −0.016 * | 0.098 *** | 0.022 *** | −0.056 *** |
| $X_{25}$ *healthcare accessibility* | −0.010 | 0.028 *** | 0.024 *** | 0.080 *** |
| $X_{26}$ *kindergarten accessibility* | −0.009 *** | 0.040 *** | 0.021 *** | 0.052 *** |
| **Policy Environment** | | | | |
| $X_{27}$*proportion of policy village* | −0.032 *** | −0.020 * | 0.016 | 0.020 |
| $X_{28}$ *proportion of reserve* | 0.022 * | −0.042 *** | 0.061 *** | −0.009 |

[1] * $p < 0.05$; ** $p < 0.01$; *** $p < 0.001$; (H) $q$ statistic $> 0.1$.

We concluded one aspect of the analysis, that is, the main influencing factors and the differences between them, by splitting the areas of study into various house-purchasing destinations. All of the factors originated from rural areas as well as rural households, so they had the most drastic impact when a farmer purchased a house in their home township or county town. Moreover, the more distant the destination, the less restrictive the rural economic environment, and the location environment grew stronger.

4.2.2. Comparison of Different Environments

Next, we analyzed the factor variation in each environment. The natural environment was the weakest of all influencing factors, which was related to the strong natural homogeneity of rural Huai'an. All factors of the settlement environment decreased with the destination outward. Of these, *per capita arable land* ($X_4$) had a stronger effect on purchases in the township and county town, suggesting that farmers who purchased locally were still influenced by their core industry and inherent occupation: cultivation. Owning land implies, on the one hand, an abundance of household assets and, on the other hand, a profit [22]. The housing environment was analyzed in the previous article, and it was the most influential of all environments. The economic environment showed a different trend, with a stronger impact when the purchase destinations are in foreign cities as more capital is required. Another small peak appeared in purchases in the county town. The population and family environment was similar to the settlement environment in that the family can be seen as a reduced settlement. The factors *household density* ($X_{16}$) and *proportion of vacant houses* ($X_{18}$) had the greatest effect when the destination was a township or a county town, suggesting that group behavior in rural societies had a strong bearing on individuals. The essence of the location environment was geographical proximity, and any economic behavior must take into account the distance factor, which has been illustrated. The influence of the public service environment was low, with all values below 0.1. Among them, *education accessibility* ($X_{24}$) was more volatile, which is due to the distribution of quality education resources, especially high schools, in metropolitan areas and counties. In short, farmers chose to purchase in a larger city because of the large public service gap [16]. The last is the policy environment, which had a weak impact. The factor *proportion of policy village* ($X_{27}$) showed a trend of increasing and then decreasing, which means that the impact of policies on purchases in the local area was increasing, but not significant for foreign destinations.

The analysis of the different environmental categories found that most influence decreased as the house purchase destination was further away, but the influence of the economic environment and the public service environment increased. This fully indicates that the purchase of houses in foreign cities may be constrained by the economic status of rural households, and there may be a disproportionate gap between Huai'an and foreign

cities in terms of public services. Furthermore, the natural environment and the policy environment could hardly influence the choice of any house purchase destination. We argue that a higher demand for housing is growing in China's less developed rural areas, creating a situation where the metropolitan area is the core and the county town is the main contributor.

### 4.2.3. Validation of Interaction Detection

Table 5 shows the results of the top 28 ranked factors that drive purchasing intensity using the interaction detector. The results reveal that interactions among the influencing factors were more significant than the effect of individual indicators on urban house purchases, and their interaction attributes were all Enhance-nonlinear and Enhance-bivariate. The interaction force decreased as the house purchase destination was further out, which could match the factor detector results. Among them, the interaction of *fiscal revenue* and *distance to metro area* ($X_{13} \cap X_{19}$) had the most significant effect on farmers purchasing houses in the township. In addition, the top-ranked *q* statistic generally interacted with *proportion of buildings* ($X_7$) in areas where the destination was a township or a county town, suggesting that the rigid demand for buildings played an important role in influencing farmers to purchase houses locally. For destinations in metropolitan areas, *distance to metro area* ($X_{19}$) became the most significant and only factor with strong interaction. Finally, the result of purchases in foreign cities tells us that *distance to metro area* ($X_{19}$) and *distance to county town* ($X_{20}$) had relatively high interaction influences, indicating that location was an important factor in remote heterogeneous urbanization.

**Table 5.** Results of the interaction detector.

| Model 1: In Township | | Model 2: In County Town | | Model 3: In Metropolitan Area | | Model 4: In Foreign Cities | |
|---|---|---|---|---|---|---|---|
| Inter. | q Statistic | Inter. | q Statistic | Inter. | q Statistic | Inter. | q Statistic |
| $X_{13} \cap X_{19}$ | 0.432 * | $X_{19} \cap X_{25}$ | 0.404 | $X_{13} \cap X_{19}$ | 0.350 | $X_{19} \cap X_{20}$ | 0.330 |
| $X_7 \cap X_{13}$ | 0.432 | $X_{19} \cap X_{20}$ | 0.394 | $X_{19} \cap X_{28}$ | 0.344 | $X_{13} \cap X_{20}$ | 0.315 |
| $X_7 \cap X_{24}$ | 0.402 | $X_7 \cap X_{13}$ | 0.389 | $X_{18} \cap X_{19}$ | 0.330 | $X_{13} \cap X_{19}$ | 0.314 |
| $X_{19} \cap X_{25}$ | 0.395 | $X_{13} \cap X_{19}$ | 0.375 | $X_1 \cap X_{19}$ | 0.321 | $X_{19} \cap X_{25}$ | 0.299 |
| $X_7 \cap X_{18}$ | 0.383 | $X_7 \cap X_{24}$ | 0.362 | $X_{14} \cap X_{19}$ | 0.319 | $X_7 \cap X_{13}$ | 0.277 |
| $X_8 \cap X_{13}$ | 0.383 | $X_7 \cap X_{19}$ | 0.358 | $X_{19} \cap X_{25}$ | 0.311 | $X_{13} \cap X_{25}$ | 0.273 |
| $X_{13} \cap X_{18}$ | 0.382 | $X_{16} \cap X_{19}$ | 0.349 | $X_{15} \cap X_{19}$ | 0.309 | $X_{13} \cap X_{24}$ | 0.268 |
| $X_8 \cap X_{18}$ | 0.381 | $X_{19} \cap X_{24}$ | 0.330 | $X_{19} \cap X_{20}$ | 0.304 | $X_{19} \cap X_{24}$ | 0.265 |
| $X_{13} \cap X_{16}$ | 0.378 | $X_7 \cap X_{21}$ | 0.325 | $X_{19} \cap X_{21}$ | 0.303 | $X_9 \cap X_{13}$ | 0.261 |
| $X_4 \cap X_{18}$ | 0.373 | $X_7 \cap X_{16}$ | 0.325 | $X_2 \cap X_{19}$ | 0.302 | $X_{11} \cap X_{19}$ | 0.258 |
| $X_{19} \cap X_{24}$ | 0.373 | $X_7 \cap X_{20}$ | 0.308 | $X_5 \cap X_{19}$ | 0.302 | $X_{13} \cap X_{26}$ | 0.257 |
| $X_{18} \cap X_{22}$ | 0.370 | $X_7 \cap X_{18}$ | 0.306 | $X_{19} \cap X_{23}$ | 0.299 | $X_{19} \cap X_{26}$ | 0.252 |
| $X_4 \cap X_{13}$ | 0.370 | $X_7 \cap X_{10}$ | 0.305 | $X_{11} \cap X_{19}$ | 0.298 | $X_{11} \cap X_{13}$ | 0.252 |
| $X_{16} \cap X_{18}$ | 0.362 | $X_7 \cap X_{25}$ | 0.302 | $X_4 \cap X_{19}$ | 0.297 | $X_{19} \cap X_{21}$ | 0.251 |
| $X_9 \cap X_{13}$ | 0.358 | $X_7 \cap X_{17}$ | 0.298 | $X_{19} \cap X_{27}$ | 0.297 | $X_9 \cap X_{16}$ | 0.240 |
| $X_7 \cap X_{22}$ | 0.355 | $X_7 \cap X_{14}$ | 0.297 | $X_3 \cap X_{19}$ | 0.294 | $X_{13} \cap X_{14}$ | 0.237 |
| $X_7 \cap X_{16}$ | 0.355 | $X_{18} \cap X_{19}$ | 0.297 | $X_{17} \cap X_{19}$ | 0.292 | $X_{11} \cap X_{20}$ | 0.236 |
| $X_4 \cap X_7$ | 0.352 | $X_1 \cap X_7$ | 0.297 | $X_{16} \cap X_{19}$ | 0.291 | $X_7 \cap X_{19}$ | 0.234 |
| $X_7 \cap X_{19}$ | 0.350 | $X_7 \cap X_{11}$ | 0.295 | $X_{19} \cap X_{24}$ | 0.290 | $X_{13} \cap X_{21}$ | 0.234 |
| $X_{16} \cap X_{19}$ | 0.346 | $X_4 \cap X_7$ | 0.295 | $X_9 \cap X_{19}$ | 0.289 | $X_9 \cap X_{19}$ | 0.231 |
| $X_4 \cap X_9$ | 0.338 | $X_7 \cap X_{22}$ | 0.293 | $X_{19} \cap X_{26}$ | 0.287 | $X_7 \cap X_{11}$ | 0.227 |
| $X_6 \cap X_7$ | 0.338 | $X_6 \cap X_7$ | 0.293 | $X_8 \cap X_{19}$ | 0.286 | $X_{11} \cap X_{25}$ | 0.226 |
| $X_9 \cap X_{16}$ | 0.336 | $X_3 \cap X_7$ | 0.288 | $X_7 \cap X_{19}$ | 0.285 | $X_{13} \cap X_{16}$ | 0.224 |
| $X_7 \cap X_9$ | 0.333 | $X_7 \cap X_{15}$ | 0.286 | $X_6 \cap X_{19}$ | 0.285 | $X_8 \cap X_{19}$ | 0.224 |
| $X_7 \cap X_{23}$ | 0.331 | $X_5 \cap X_7$ | 0.285 | $X_{10} \cap X_{19}$ | 0.281 | $X_{17} \cap X_{19}$ | 0.223 |
| $X_7 \cap X_{17}$ | 0.331 | $X_7 \cap X_{26}$ | 0.283 | $X_{12} \cap X_{19}$ | 0.275 | $X_8 \cap X_{13}$ | 0.223 |
| $X_7 \cap X_{25}$ | 0.330 | $X_{19} \cap X_{21}$ | 0.283 | $X_{19} \cap X_{22}$ | 0.275 | $X_{24} \cap X_{25}$ | 0.223 |
| $X_1 \cap X_7$ | 0.329 | $X_7 \cap X_8$ | 0.282 | $X_7 \cap X_{13}$ | 0.252 | $X_{13} \cap X_{18}$ | 0.222 |

* Single underlined are Enhance-nonlinear, the others are Enhance-bifactor.

## 5. Conclusions and Discussion

### 5.1. Conclusions

In China, one of the main reasons for investment in infrastructure and transportation has derived from rural–urban migration, making it important to study these factors in a more disaggregated way. On the basis of the unique survey data of rural areas in Huai'an, in this paper, we categorized the house purchase behavior of amphibious farmers in urban areas from the perspective of the rural environment. The results of the descriptive statistics and spatial analysis show that different types of house purchases were highly variable and had distinct correlations with different environments.

The average urban house purchase ratio in rural Huai'an was 45.96%, indicating that the study population was a large group that cannot be ignored. Specifically, there were large disparities in house purchases across destinations. The county town became the most important destination for rural households to purchase houses (18.78%), followed by the township (14.63%). In addition, the metropolitan area of Huai'an (4.82%), a sub-developed region, was significantly less attractive than foreign cities (7.74%). In terms of spatial distribution, the areas with high purchase rates in the hometown were concentrated in the township with strong development levels, such as Maba Town; the areas with high purchase rates in the county town were concentrated around the central built-up areas of Jinhu County, Hongze District, and Lianshui County, which were characterized by a single-core urban system; villages on the border of Huai'an's central urban area tended to purchase in the metropolitan area; and villages on the edge of Huai'an purchased in foreign cities. In short, the level of economic development was the fundamental factor determining farmers' housing choices.

The heterogeneity of different environmental drivers was also evident. In this study, we found that all types of rural living environment factors had a more significant impact when farmers were purchasing houses nearby (in the township and county town). Among them, the housing environment, the settlement environment, and the population and family environment were the main environmental impact elements, and the factor with the largest impact was *proportion of buildings*. Moreover, the location environment was the most critical factor influencing house-purchasing in an exotic location (metropolitan area and foreign cities), as well as the economic environment. However, the natural environment and the policy environment had little influence on any farmer's house-purchasing destination. Finally, we used the interaction detector and found that the post-interaction influences were both Enhance-nonlinear and Enhance-bivariate. Among them, *distance to metro area* had a stronger interactive effect, and interaction force decreased as the house purchase destination was further away, which reconfirmed the conclusion of the factor detector.

### 5.2. Discussion

#### 5.2.1. Policy Recommendations

In addition to a few megacities (e.g., Shanghai and Beijing) that are impacting the world's cities, there is a handful of emerging regional dominant megacities (e.g., Nanjing), a large number of prefecture-level cities (e.g., Huai'an) that are in the midst of high-quality transition, and a very large number of fast-growing small cities and towns (e.g., Lianshui County). However, the developing small and medium-sized cities hidden around the developed cities are often overlooked. This paper argues that the mechanism inherent in population mobility is differential poverty caused by the rural environmental gap and suggests that attention should be paid to the original living environment of rural households. The conclusion is that more farmers are choosing to purchase a house in their hometown or metropolitan area rather than in Shanghai or Nanjing, which is an important embodiment of China's in situ urbanization and New Urbanization Strategy. The following are some policy recommendations.

First, from a national perspective, we should continue to pay more attention to small and medium-sized cities and inject preferential policies to prevent the decay of the national urban system. It is urgent to change the urbanization mindset and policy orientation that

excessively relies on farmers' house-ownership and to consider the housing needs of both the moving-out (e.g., counties) and moving-in (e.g., provincial capitals) areas in the goal of urban–rural integration. If the rural population disproportionately flows to megacities, it will be detrimental to China's food security and sustainable territorial development. As Opinions on Promoting Urbanization with the County as an Important Carrier [65] issued in May 2022 is a very meaningful policy, more similar ones should be proposed.

Second, we call on the government to develop a more differentiated policy to treat "new citizens" from rural areas in a friendly manner by offering some housing concessions and welfare. For example, Huai'an city and county should exempt the deed tax on first house purchases for rural households or provide substantial low-rent houses, which would facilitate in situ urbanization rather than population loss. In addition, restrictions on hukou, medical care, and education should be relaxed. Except for a few megacities, hukou restrictions should be completely removed; education for farmers' children and medical insurance should be treated the same as those for citizens.

Third, for those who prefer to stay in the countryside, there should be significant improvements in housing and public services. We strongly recommend the promotion of countryside centralized residence by category, which would allow for the replacement of houses and the improvement of the environment. In addition, rural industries should be modernized and intensified, so that employment and income issues can be addressed. Only then will farmers be able to choose their destinations more rationally and freely, high-quality urbanization will flourish, and "new farmers" will have a brighter life.

### 5.2.2. Limitations and Future Directions

This paper still has some research limitations. Our four classifications may be somewhat broad and limited to Huai'an, providing only limited geographical explanations and references. Although we tried to identify a more generalized environmental research framework, for the time being, we did not consider and quantify the local customs. What is more, our data included a sample of rural households that had purchased a house in foreign cities smaller than Huai'an, although there were few such households in the study. Last but not least, pull influences were missing due to the unavailability of the specific city where the house was purchased. Therefore, we believe that a comprehensive push–pull framework should be developed.

Next, the concept of amphibious farmers needs to be further investigated. In terms of technology, big data (e.g., cell phone signaling) of rural–urban population migration should be systematically portrayed to help derive the deep-seated mechanism of population flows. Finally, the currently prevailing trend of urban–rural network connection is likely to be the future pattern of population mobility, and multi-flow research should be strengthened.

**Author Contributions:** Conceptualization, C.W., Z.P. and C.G.C.; methodology, C.W.; software, Z.P.; validation, C.W., C.G.C. and Z.P.; formal analysis, Z.P.; investigation, C.W.; resources, C.G.C.; data curation, C.W. and C.G.C.; writing—original draft preparation, C.W. and Z.P.; writing—review and editing, C.G.C.; visualization, C.W.; supervision, C.G.C.; project administration, C.G.C.; funding acquisition, C.W. All authors have read and agreed to the published version of the manuscript.

**Funding:** This research was supported by the National Natural Science Foundation of China (52078237); and the National Research Foundation of Korea grant, which is funded by the Government of South Korea (NRF-2020R1A2C1008509).

**Data Availability Statement:** The sources and preprocessing of data are in Section 3.1. The data produced in this study are available on request from the corresponding author.

**Conflicts of Interest:** The authors declare no conflict of interest.

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
