# Peer review of "Township, County Town, Metropolitan Area, or Foreign Cities? Evidence from House Purchases by Rural Households in China"

_land, doi:10.3390/land12051038_

Round 1

Reviewer 1 Report

The topic selection of the paper is quite interesting, and the argumentation process is also quite thorough. Overall, the paper is worth publishing. However, there are also some issues with the paper.

 1.     The topic selection of the paper is quite interesting, and the argumentation process is also quite thorough. Overall, the paper is worth publishing. However, there are also some issues with the paper.

 2.     Line 222, What is the basis for dividing the impact of village geographical environment on rural housing purchase into eight categories? Is there a basis for proposing specific indicators under each category?

 3.     Line 376, Table 3 has an extra dot.

 4.     The specific meanings of "metropolitan area" and "foreign city" are suggested to be explained.

 5.     The conclusion section is too simplistic and has not been further refined based on the research results. For example, the meaning expressed in the second paragraph of the conclusion should be that the level of economic development (which means more employment opportunities and higher salary levels) is the fundamental factor determining farmers' housing choices.

 6.     The policy recommendations do not seem to correspond with the research results, and the correlation with the analysis results in this article is not strong enough.

Author Response

Dear reviewer:

We are honored to receive your appreciation, and your suggestions will greatly contribute to the quality of our article.

Our point-to-point responses:

A1: Thanks!

A2: We did want to include as many influencing factors as possible and to construct a comprehensive environmental framework. As we wrote in the article, “previous studies and field surveys in rural Huai'an” is the basis for dividing into eight categories. To fill the gap, we added several references to illustrate this issue, but were only marked yellow because the revision in Word would cause collapse (Endnote's full-text order adjustment). In addition, previous linear regression methods had covariance and could not incorporate so many factors; the GeoDetector we used could overcome that problem. (3.2.2.)

A3: Revised.

A4: We added 3.2.1. to explain this confusion. The metropolitan area only represents the central urban area of Huai'an ("ShiQu"), and the foreign city represents any other city ("WaiDi").

A5: Thanks for this comment. We have done some complication and summarization, and any further revisions can be implemented. (5.1.)

A6: We have made more specific and targeted policy recommendations with the results, and any further revisions can be implemented if you propose. (5.2.)

There were many slight changes throughout the manuscript, which we checked in full.

Finally, your comments are very helpful to improve the quality of our article, thanks again!

Best wishes!

Reviewer 2 Report

The topic addressed by the research exposes the existing problems in the migratory behavior of farmers to the cities in China, which is causing the aging of the rural population, as well as problems in the real estate markets and in the public services that receive the migrant population.

The authors do not indicate the research question, hypotheses or research objective. In the document, different objectives appear in the abstract: 

- "to study the differences in housing purchased by rural households in terms of destinations in different classes of cities and the factors influencing these purchases were analyzed with GeoDetector."

In the introduction.

- "it is important to identify the house purchase characteristics of rural households to optimize the layout of public facilities and promote the coordinated development of urban and rural areas.”

Or in section 3.2.1 Selection of variables.

- "discusses is the changes in the living conditions of rural residents, and the housing environment in the original villages must be the main influencing factor.”

Therefore, it is recommended that the authors, after the description of the topic, clearly state the problem to be analyzed in the paper, describe the research objectives (they should be the same throughout the paper), the research questions and the methodology used.

The methodology section is clearly stated and presented, while the results are also clearly stated. Regarding section 5. Conclusions and discussion, the authors have been able to define the policy implications. At the same time it is suggested that the authors change the first sentence to:

"In China, one of the main reasons for investment in infrastructure and transportation has derived from rural-urban migration."

Author Response

Dear reviewer:

We are honored to receive your appreciation, and your suggestions will greatly contribute to the quality of our article.

Our point-to-point responses:

A1: You are correct in pointing out the problem. We have aligned all research issues and objectives in this manuscript.

A2: Thanks for this suggestion. We've added that sentence and completely deepened the section 5.

There were many slight changes throughout the manuscript, which we checked in full.

Finally, your comments are very helpful to improve the quality of our article, thanks again!

Best wishes!

Reviewer 3 Report

This is an interesting study, and the authors have collected a unique dataset to analyse and represent the results in a structured way. Overall the information presented represents valuable information regarding the preferences of rural households on the housing purchase.

Key critical points are:

1.      English needs significant improvement while I understand the core message. The texts are often vague and long-winded. For example, the whole introduction section can be simplified and more compact. The authors should have the manuscript edited by an English professional.

2.      The storyline of the literature review is quite confusing. The authors illustrated a lot of theories and brought up different indicators influencing rural-urban migration, but they did not convey valuable and clear information to the audience. I suggest the authors re-organise this section and put focus on the Chinese context, and also think about the key message that could contribute to the following section. An encyclopaedia is not necessary. Besides, the unclear expression gives me a feeling that several statements are just authors’ assumptions rather than facts. Please recheck the whole chapter.

3.      In several instances, I also suggested citing more relevant and recent literature. It lacks solid reference support on study area and data throughout section 3.1. Reference is also needed for the 14th Five-Year Plan.

4.      I still cannot distinguish very clearly between county-town, township, foreign cities, and metropolitan areas. What do they mean? I suggest the authors have definitions of these specific terms in the context. Does a foreign city mean a city outside of China mainland or a city outside of the province? I think it is very important to define the range for the audience. For example, you can bring the Chinese pinyin or the character into the context for better explanations.

5.      There are a lot of inferences and subjective estimations of why this and why that in Chapter 4. Please add corresponding citations to support the statements. Otherwise, please delete the unreasonable ones.

6.      Line 520-531. The paragraph can be analysed more in-depth and less superficial. Now it looks superficial and not informative. The same advice applies to section 5.2.2. Please re-organise them.

Given these shortcomings, the manuscript requires major revisions. Additional suggestions are made as below.

1.      P2, line 61-63. I suggest to cite and introduce several policies and regulations to give the audience a brief background understanding.

2.      P2, line 68-69. The language is very vague, how do you define adequate, high-grade and good? Is that what illustrated in the references?

3.      P2, line 86-91. Please indicate the number sequences for each chapter.

4.      Line 167-175, should this paragraph be placed in the end of introduction?

5.      Please put Figure 1 after the corresponding texts.

6.      Line 508, nearing saturation? Any reference support or your personal assumption?

7.      Line 516-518. The conclusion is that more farmers are choosing to purchase a house in their home town or the metropolitan area rather than Shanghai or Nanjing, which is a milestone achievement of China's in-situ urbanization and New Urbanization Strategy. This conclusion is consensual – how can everybody afford a apartment in Shanghai or Beijing? How does that represent a milestone achievement of China’s urbanisation strategy?

Author Response

Dear reviewer:

We are honored to receive your appreciation, and your suggestions will greatly contribute to the quality of our article.

Our point-to-point responses:

A1: Sorry for our poor language. We performed full-text checking and extensive rewriting, and also used MDPI's English Editing service. Proof of English editing was submitted to the editor.

A2: There are indeed huge problems with the literature review. We have regrouped the literature review into two sections: "Homeownership: Environmental Inequality" and "From Village to City: Story of China's Farmers". Also, we clarified the logic, explained the purpose, and kept it close to the themes of the article. (Section 2.)

A3: Revised. Some references were added and replaced, but were only marked yellow because the revision mode in Word would cause collapse (Endnote's full-text order adjustment).

A4: We added the 3.2.1. to explain this confusion.

A5: Revised. Some sentences have been modified and some literatures have been added. (Section 4.)

A6: We have made more specific and targeted policy recommendations, and any further revisions can be implemented if you propose. We have also made our best efforts to revise 5.2.2.

Additional A1: Revised.

Additional A2: Sorry, we removed the vague rhetoric and only used better, and modified the references.

Additional A3: Revised.

Additional A4: Revised. Because chapters 1 & 2 have been overhauled.

Additional A5: Revised.

Additional A6: Revised.

Additional A7: Sorry, that was not our intent. We just wanted to highlight the comparison because Huai'an is not a big city either. This sentence was revised.

There were many slight changes throughout the manuscript, which we checked in full.

Finally, your comments are very helpful to improve the quality of our article, thanks again!

Best wishes!

Reviewer 4 Report

The focus of this article is to analyze the phenomenon of population migration in rural areas. Four types of urbanization caused by population migration in Huaian City are analyzed with GeoDetector. The following suggestions and comments are provided for the author as a reference for revision.

1. Line 194-201. This paragraph is explaining the seriousness of population loss in Huaian City, and it would be more convincing to provide evidence from literature or related research reports.

2. Line 316. Proposing that two variables act together, does the implication mean that the two variables are multiplied together? What is the rationale for discussing two-variable interaction effects in this paper?

3. Table 3 and Figure 3 should be the spatial distribution of house purchase rates rather than the house purchases.

4. Is the definition of slope in Table 1 degrees or radian?

5. The variables used in this paper are all “push” factors in the rural areas of Huaian City. It is suggested that follow-up research can moderately add the factor of "pull" in the immigrant areas.

Author Response

Dear reviewer:

We are honored to receive your appreciation, and your suggestions will greatly contribute to the quality of our article.

Our point-to-point responses:

A1: We have tried our best to add some literature on rural population loss in Huai'an or the regional inequality and polarization of Jiangsu province, in the fixed line.

A2: Yes, the Interaction GeoDetector can be interpreted as 2 factors multiplied by each other. The reason we used is to deepen the results of the baseline regression to reinforce the role of the purchase factor. Some sentences were added in the paper.

A3: Revised.

A4: The definition of slope is degrees, not radian. We have added a description “degree” in Table 1.

A5: Thank you for your suggestion! The crucial reason is that our data are in administrative villages but not in individuals, so we can only get “rates”, not the corresponding “pull” of immigrant cities. In any case, we will improve it in our follow-up research.

There were many slight changes throughout the manuscript, which we checked in full.

Finally, your comments are very helpful to improve the quality of our article, thanks again!

Best wishes!